# Ozone Pollution, Oxidative Stress, Regulatory T Cells and Antioxidants

**DOI:** 10.3390/antiox11081553

**Published:** 2022-08-11

**Authors:** Selva Rivas-Arancibia, Eduardo Hernández-Orozco, Erika Rodríguez-Martínez, Marlen Valdés-Fuentes, Vanessa Cornejo-Trejo, Nelva Pérez-Pacheco, Claudia Dorado-Martínez, Diana Zequeida-Carmona, Isaac Espinosa-Caleti

**Affiliations:** Facultad de Medicina, Universidad Nacional Autónoma de México, Ciudad de México 04510, Mexico

**Keywords:** ozone pollution, oxidative stress state, regulatory T cells, inflammation, antioxidant

## Abstract

Ozone pollution, is a serious health problem worldwide. Repeated exposure to low ozone doses causes a loss of regulation of the oxidation–reduction systems, and also induces a chronic state of oxidative stress. This fact is of special importance for the regulation of different systems including the immune system and the inflammatory response. In addition, the oxidation–reduction balance modulates the homeostasis of these and other complex systems such as metabolism, survival capacity, cell renewal, and brain repair, etc. Likewise, it has been widely demonstrated that in chronic degenerative diseases, an alteration in the oxide-reduction balance is present, and this alteration causes a chronic loss in the regulation of the immune response and the inflammatory process. This is because reactive oxygen species disrupt different signaling pathways. Such pathways are related to the role of regulatory T cells (Treg) in inflammation. This causes an increase in chronic deterioration in the degenerative disease over time. The objective of this review was to study the relationship between environmental ozone pollution, the chronic state of oxidative stress and its effect on Treg cells, which causes the loss of regulation in the inflammatory response as well as the role played by antioxidant systems in various pathologies.

## 1. Introduction

As the population ages, non-infectious chronic-degenerative diseases also increase in the population, which are generally characterized by presenting a chronic state of oxidative stress, and the loss of regulation of the inflammatory response, which changes and increases over time. Both the chronic increase in reactive oxygen species (ROS) and the alteration in inflammation are key factors during the progress and deterioration of chronic disease [1]. Exposure to environmental pollutants such as ozone leads to the production of ROS secondary to their exposure in the interstitial fluid, these induce the production of proinflammatory cytokines, both in the brain and in the lungs. The air that we breathe with low doses of ozone produces ROS when in contact with the olfactory receptors, passes to the olfactory bulb, and reaches different brain structures. When inspired, ozone also reaches the respiratory tract where ROS are formed. When these cannot be compensated by antioxidant defenses, they are distributed throughout the circulation and reach the entire body, causing a pro-oxidant environment and raising pro-inflammatory cytokines [2] (Figure 1). The immune response, therefore, is an increase in the Th1–Th17 response and a decrease in the Th2 response [3]. Oxidative signals participate in the modulation of the immune system, Treg cells respond to oxidative changes in the environment by increasing or decreasing their number and are key in regulating the inflammatory response. On the other hand, exposure to ozone causes a state of chronic oxidative stress. This loss of oxide-reduction balance induces a loss of regulation of the inflammatory response on the immune system and alters the response of Treg cells.

In murine models, we have reported that exposure to low doses of ozone (0.25 parts per million) in healthy rats causes alterations at the molecular, cellular, and systemic levels, which, depending on the exposure time, produces a process of progressive neurodegeneration similar to what happens in neurodegenerative diseases such as Parkinson’s and Alzheimer’s disease [4,5]. Furthermore, at the molecular level, we found an increase in prooxidants, alteration of endogenous antioxidant systems, and oxidative alteration in the metabolism of neurotransmitters such as dopamine [6]. Additionally, at the cellular level, the changes were characterized by inflammation, edema of the endoplasmic reticulum and mitochondria, cell membrane disruption, intracellular amyloid beta 1–42 accumulation, the formation of extracellular amyloid plaques and neuronal death, gliosis, and phagocytic microglia [2]. These changes are accompanied by an increase in the Th1–Th17 response and a significant decrease in the Th2 response, as has also been found in many degenerative pathologies. On the other hand, despite the fact that there are many studies on the modulation of the redox system and the role that regulatory T cells play both in the regulation of the immune system and in the inflammatory response in different diseases, much remains to be elucidated as well as the role that antioxidant systems may be playing in said regulation and in future therapies.

## 2. Ozone Pollution

In the troposphere, ozone (O_3_) acts as a pollutant, since it is produced when nitrogen oxides and volatile organic compounds from sources such as fuel burning react in the presence of sunlight, generating O_3_. It causes irritation and inflammation of the eyes, nose, throat, and lower respiratory tract, causing coughing, pain, and scratchy throat. O_3_ is a secondary pollutant, since it is generated from other gases such as nitrous oxide and nitric oxide in conjunction with volatile organic compounds and UV light. Both nitrous oxide and nitric oxide are pollutants generated by the combustion of hydrocarbons in automobile engines and in various industrial processes [7]. The molecular conformation of O_3_ gives it highly reactive and oxidizing properties, since being breathed repeatedly significantly increases the ROS in the body; however, antioxidant systems do not respond adequately to low and repeated doses of ozone, which leads to an imbalance in oxidation–reduction reactions (redox) [2].

The loss of redox balance due to chronic exposure to ozone produces the oxidation of lipids, DNA, and proteins as well as energy failure due to adenosine triphosphate (ATP) deficiency, epigenetic alterations, and finally metabolic alterations that lead to cell death [5,8]. ROS have a crucial role in the development of neurodegenerative diseases such as Alzheimer’s disease [9], Parkinson’s disease [10], multiple sclerosis [11], amyotrophic lateral sclerosis, and Huntington’s disease [12,13]. In addition, the state of oxidative stress is present in the development of degenerative diseases, autoimmune diseases, cancer, cardiovascular diseases, diabetes, obesity, etc. The thymic epithelial cell, together with the thymic and induced Treg cells, play a very important role in the regulation of the inflammatory response. Immunological tolerance is responsible for protecting our body by inhibiting the activation of responses to self-antigens; among the most important mechanisms we have are regulatory T lymphocytes (Figure 2). For the maturation of regulatory T lymphocytes to take place, a series of cellular interactions and different regulatory molecules are required including reactive oxygen species (ROS), which play a key role in this process [14].

## 3. T Regulatory Cells, Inflammation, and Peripheral Inflammation

Thymic differentiation of Foxp3+ Treg cells requires three main types of signals: (a) ligand-dependent recognition of T-cell antigen receptor (TCR) peptide/MHC-II (pMHC-II); (b) triggered CD28-dependent signaling by costimulatory ligands CD80 or CD86; and (c) cytokine signaling triggered by IL-2 detection. Treg cell differentiation is highly dependent on ligand presentation through three types of antigen-presenting cells (APCs). During maturation in the thymus, thymocytes that exhibit overt reactivity to self-pMHC-II ligands may undergo Treg cell differentiation or clonal deletion [15]. According to the classification of Li (2016) and Savage et al. (2020), there are a series of factors that determine the alternative fates of Treg cells: stochastic, TCR/pMHC-II binding properties, ligand density, type of APC, maturational stage, and age-dependent effects [15,16]. It is not clear whether iTregs and pTregs are similar populations that use the same suppressive mechanisms (Figure 2). Most of the suppressive activity of freshly isolated Foxp3+ T cells is mediated by nTregs. However, little is known about the mechanisms of action of Tregs in vivo [17].

## 4. Treg Cells and Inflammation

The immune system is finely regulated to attack foreign antigens and tolerate self-antigens. Treg cells are a special subpopulation of immunosuppressive T cells that maintain self-tolerance, inhibit autoimmunity, and act as critical negative regulators of inflammation in various physiological states such as pregnancy or aging, and pathological states including autoimmunity, injury, and degeneration of the nervous system (Figure 3). Treg cells restrain immune responses through the elaboration of suppressor function that is dependent on Foxp3 expression. The deregulation of any of these processes can have serious consequences [18].

Despite a critical role of Treg cells in maintaining lympho-myeloid homeostasis, it remains unclear when multiple mechanisms of Treg cell-mediated suppression are operating in vivo and how redundant such mechanisms might be. Classically, the functions of Treg cells have been described in CD4+ T cells, but other immune cells such as some CD8+ T cells, gamma delta T cells (γδT), or cytotoxic (natural killer or NKT) cells also have the ability to modulate the response immune. The main non-lymphoid tissues where Treg cells can be found are visceral adipose tissue, intestine, skin, and muscle. In these four tissues, Tregs are important regulators of inflammation and fibrosis and contribute to tissue repair [19]. Treg cells finely and dynamically orchestrate their response to different physiological or pathological scenarios.

Treg cells participate in processes in which they adapt to the environment to maintain tissue homeostasis. Treg cells are present in healthy tissues and, after tissue injury, promote tissue regeneration in a manner dependent on the transmembrane glycoprotein of the epidermoid growth factor family or protein amphiregulin or AREG [20]. In conditions such as pregnancy, transplants, or the presence of the microbiota, tolerance to foreign antigens is required, for which Treg activity is essential (Figure 3). If Treg cell activity is too low, there may be a failure of self-tolerance, leading to the development of autoimmune diseases. On the other hand, if Treg cells are hyperactive, they can favor the progression of malignant neoplasms.

### Treg Cells and Infections

Low responsiveness and reduced proliferation of virus-specific T cells during chronic viral infection are associated with Treg cell expansion. In models of acute and chronic murine retroviral infection, Treg cell depletion decreases viral load and restores virus-specific cytotoxic CD8+ T cell activity. In tuberculosis, the role of Treg cells depends on the stage of the disease; Treg cells expand and delay immune responses in the early phases, but counter-regulate excessive inflammation later in the chronic phase [21]. In a mouse model of chronic Friend retrovirus (FV) infection, vaccination with a calcium phosphate nanoparticle adjuvant, which efficiently reactivated CD8+ T cells, in combination with transient Treg cell ablation, enhanced antiviral immunity [22].

## 5. Regulatory T Cells and Disease

### 5.1. Autoimmunity (Target)

The immune system has the ability to distinguish foreign agents from its own through different mechanisms, which together is known as immunological tolerance [23]. When this process is altered, the immune system will begin a reaction against its own antigens, generating autoimmunity, causing inflammatory responses that characterize the more than 80 clinical presentations that make up autoimmune diseases [24].

There is no single pathway for the generation of autoimmunity, but rather a complex system of interconnected pathways, in which the two mechanisms of immune tolerance must be compromised: central and peripheral, accompanied by genetic predisposition and environmental factors [24]. It is the thymus where central tolerance is carried out, with elimination by the negative selection of lymphocytes; if these bind avidly to the MHC–peptide complex itself, or to the tissue-restricted antigens (TRA) presented by the medullary thymic epithelial cell (mTEC), they will die by apoptosis. The autoimmune regulator AIRE that nTECs present when mutated can generate systemic autoimmune diseases such as autoimmune polyendocrinopathy-candidiasis-ectodermal dystrophy syndrome (APECED) [23].

Central tolerance is not perfect and relies on peripheral tolerance to prevent autoimmunity. It has also been shown that in C57BL/6 mice, AIRE deficiency was not sufficient to cause severe disease because they were also regulated by PD-1-mediated peripheral tolerance, whereas AIRE and PD-1 deficient mice suffered a rapid, multiorgan, and fatal autoimmune disease [25]. Despite Treg cell suppression and other mechanisms of peripheral tolerance such as the presence of inhibitory molecules and anergy in genetically predisposed individuals, the presence of tissue damage, inflammation, and the presentation of self-antigens or microbial mimetics can cause a breakdown in tolerance [24].

### 5.2. Oxidative Stress

Usually, air pollution has been shown to be a key factor in the generation of autoimmune diseases and their increased severity (Figure 4) [26,27,28]. In another study, it was shown that among the different environmental pollutants, ozone was the only one to remain constant regardless of the season of the year, in relation to the risk of relapse in multiple sclerosis [29].

The role of oxidative stress has been shown in many autoimmune diseases such as systemic sclerosis, myasthenia gravis, dermatomyositis, vitiligo, and Hashimoto’s thyroiditis, among others [30,31,32,33,34]. In a recent study, the levels of ROS in peripheral blood neutrophils of patients with rheumatoid arthritis (RA) were evaluated, which, in addition to being elevated, were associated with the severity of the disease [35]. In 2015, a study in mice with CD4+ T lymphocytes epigenetically modified by oxidative stress induced the production of anti-dsDNA antibodies and glomerulonephritis, demonstrating that oxidative stress can contribute to lupus disease by inhibiting the signaling of the mitogen-activated pathway, protein kinase (ERK) on T cells, leading to DNA demethylation, the upregulation of immune genes, and autoreactivity [36].

In the pathophysiology of psoriasis, a redox imbalance is found in granulocytes and plasma; an increase in nicotinamide adenine dinucleotide phosphate (NADPH) and xanthine oxidases can be found, with a decrease in antioxidants such as thioredoxin, glutathione, and vitamins C and D. In an attempt to compensate for this imbalance, a greater production of the transcription factor nuclear erythroid factor 2 (Nrf2), responsible for antioxidant transcription, was found. There was also an increase in proinflammatory markers such as NF-κβ and TNF-α, which also increases the oxidizing activity; all this will end up generating a respiratory burst with the generation of ROS [37]. Vitiligo, likewise, presents an overproduction of ROS, through the elevated activity of NADPH oxidase [38]. NADPH oxidase is not the only enzyme that has been found to be involved in excessive ROS production in AD. Myeloperoxidase is elevated in RA, and promotes oxidative stress [39]. Furthermore, in Treg-deficient mice that end up developing lethal autoimmunity, systemic activation of the transcription factor Nrf2, responsible for the transcription of antioxidants, was shown to have a good anti-inflammatory effect [40].

Another pathway of autoimmunity related to the environment is in the dysbiosis of the microbiota, which promotes damage to the mucosal–epithelial barrier, along with the translocation of bacteria and inflammatory products to the mesenteric lymph nodes, promoting the production of inflammatory cytokines, the decrease in Treg cells, and the increase in Th17 cells, ending with the production of autoantibodies [41]. Previously, only the intestinal microbiota had been considered, but now a connection between the lung microbiota and the immune system of the cerebral nervous system has been demonstrated: the microglia, which adapts its immune response capacity according to these microbial signals, influence its susceptibility to immune diseases, [42]. This may be another key point in the relationship between the pollution in the air we breathe and the development of autoimmune diseases.

### 5.3. The Role of Treg Cells in Autoimmune Diseases

Treg cells are usually found decreased in autoimmune processes such as RA and systemic lupus erythematosus (SLE) [43] and due to their immunosuppressive capacity, they have become the object of study for the development of autoimmunity [44]. Much research has focused on the mechanisms to promote the induction of these cells such as the in vivo induction of Treg cells, through a chimeric peptide of CTLA-4, a critical protein for the maintenance of immunological tolerance expressed by Treg cells, resulting in the decrease in the progression of experimental autoimmune encephalitis [45]. On the other hand, the cytokine pathways involved in the proliferation and maintenance of T cells have also been explored. However, due to its short half-life, IL-2 has not been feasible to use as a treatment. An alternative is offered with a long-lasting glycosylated IL-2 to promote the induction of Treg cells and the suppression of autoimmunity. One study showed a significant reduction in disease progression in two autoimmune models (RA and colitis induced by dextran sulfate sodium) [46].

In addition, oxidative stress has been shown to be one of the mechanisms that affects its immunosuppressive function. In patients with SLE and antiphospholipid syndrome (APS), superoxide production was increased in all T cells, and was related to a decrease in the number of Treg cells [47]. In a murine model with type 1 diabetes mellitus (DT), Tregs cells were found with an increase in oxidative stress, specifically lipid peroxidation mediated by intracellular ROS, which could be the cause of the chronic inflammatory state of DM [48]. Another study in a murine model with experimental autoimmune encephalitis found that the removal of mitochondrial ROS from Treg cells reversed DNA damage, prevented their apoptosis, and mitigated the autoimmune response by Th1 and Th17 [24]. It was recently shown that reducing ROS or increasing superoxide dismutase activity prevented apoptosis in Treg cells activated with specific antigens [49], demonstrating the fundamental role of the redox balance and correct function of Treg cells in autoimmunity.

## 6. Treg Cells and Cancer

### 6.1. Oxidative Stress and Cancer

Tumorigenesis is a process that takes place in a state of oxidative stress (Figure 5), where excess ROS can promote DNA damage and favor the generation of oncogenic mutations or can lead to cell death [48]. As a protective mechanism, cancers have genetic mutations that provide protection against ROS by increasing the endogenous antioxidant systems of tumor cells [49]. By favoring the expression of Nrf2, a defense against oxidative stress is obtained because it induces the expression of antioxidant genes necessary for the biosynthesis of NADPH-generating enzymes, a cofactor required for peroxiredoxin- and glutathione peroxidase-dependent antioxidant regulatory pathways such as the enzyme malic located in the cytosol (ME1), isocitrate dehydrogenase (IDH1), and glucose-6-phosphate-dehydrogenase (G6PD) [50,51].

However, cells that contribute to suppressing antitumor immunity such as Treg cells have a poor response to Nrf2, which leads to the death of these cells induced by ROS, but at the same time, their apoptosis collaborates as a suppressive mechanism of the antitumor response because it promotes the release of adenosine, which has an inhibitory effect on the activity of Teff lymphocytes and APCs through the adenosine A2A receptor (A2AR) pathway [52,53]. Furthermore, within the tumor microenvironment, the high amounts of ROS generated by tumor cells, hypoxia, and persistent antigenic signals lead to decreased T-cell response [17,54,55]. For example, hydrogen peroxide (H_2_O_2_) secreted by tumor-associated macrophages promotes tumorigenesis by reducing the activity of cytotoxic T cells and NK cells [49]. This suggests that ROS modulate the behavior of cancer cells and T cells, so that according to their increase or decrease in the tumor microenvironment, they can promote or slow down tumor progression [56].

### 6.2. Treg Cells and Cancer

In cancer, Treg cells present alterations in the response of the immune system [57]. The number of tumor-infiltrating Treg cells is associated with tumor progression, poor prognosis, and decreased survival rate in patients with lung, ovarian, breast, and melanoma cancer, among others [57,58]. In addition, the existence of a large number of Treg cells and a low proportion of CD8+ T cells has been linked to a poor prognosis in ovarian, breast, and gastric cancers [59].

### 6.3. Interaction between Treg Cells and the Tumor Microenvironment

CD4+FoxP3+ Treg cells differentiate through the recognition of autoantigens presented by thymic stromal cells (Figure 4). In the tumor microenvironment, there are autoantigens associated with dying tumor cells that are more likely to be recognized by Treg cells than by Teff and memory cells, since Treg cells have a high-affinity TCR [59]. Treg cells expressing chemokine receptors CCR4, CCR5, CCR10, and CXCR3 migrate to tumor sites through chemotaxis, induced by a gradient of chemokines released by tumor-associated immune cells such as tumor-associated macrophages (TAMs) and suppressor cells of myeloid origin, myeloid-derived suppressor cells (MDSC) that produce various chemokines such as C-C chemokine ligands 17, 22, 5, 6, or 28 (CCL17, CCL22, CCL5, CCL6, or CCL28) according to the tumor or its location [57,60]. In addition, tumor and immune cells in the tumor microenvironment are capable of generating cytokines that favor the increase in Treg cells such as TGF-β, which is a necessary mediator for the differentiation and survival of Treg cells [59].

### 6.4. Suppression of Antitumor Response

Treg cells are capable of suppressing the immune response to tumors through various mechanisms, for example, through the expression of the alpha subunit of interleukin 2 (IL-2αR or CD25), Treg cells generate a decrease in the availability of the interleukin for Teff cells, since it has a high affinity for this cytokine [61,62]. Furthermore, IL-2 expression occurs during the intermediate differentiation stage of Teffy cells and ceases when they fully differentiate [63]. Therefore, Treg cells, by producing adenosine and cytokines such as TGF-β, IL-10, and IL-35, are capable of suppressing the activation and complete differentiation of Teff cells to maintain the production of IL-2 necessary for their survival [64]. Furthermore, the production of granzyme B and perforin by Treg cells is another mechanism of suppression of the immune response, since it generates the cytolysis of Teff cells [61]. On the other hand, Treg cells are capable of affecting the maturation and function of APCs through the expression of CTLA-4, which binds to the CD80 and CD86 molecules with greater affinity than to the CD28 stimulatory molecule, which induces an inhibitory signaling in APCs [65,66].

### 6.5. Treg Cells as a Therapeutic Target against Cancer

Attention has focused on strategies directed against the mechanisms of immunosuppression generated by Treg cells; however, it is important to highlight that it is difficult for the same therapy to be effective for the treatment of different tumors. Therefore, it is possible that the use of combined therapies can lead to a favorable outcome [62].

## 7. Treg Cells and Transplantation

T cells are largely responsible for the rejection or tolerance of transplanted organs due to their central role as mediators of the immune response. However, Treg cells are key to the survival of the transplanted organ and, due to their importance, new therapeutic approaches are being investigated for these cells (Figure 6).

### Treg Cells in Tolerance

Treg cells have gained much interest for the tolerance and stabilization of the post-transplanted organ, since one of the main objectives is to achieve long-term tolerance without the need to use immunosuppressive therapy. Various proposals have been described for this such as the adoptive therapy of cells, which consists of administering T cells after hematopoietic progenitor cell (HPC) transplantation, which are extracted from the patient to be later modified in order to accelerate immune recovery. This must occur early to avoid the cytokine storm in the first 2 weeks after the transplant. This treatment has been found to be widely suitable in the haploidentical hematopoietic stem cell transplantation (HSCT) of patients who do not have compatible siblings or donors, thus avoiding graft-versus-host disease. Currently, the treatment to prevent graft-versus-host disease consists of administering doses of chemotherapeutic agents to the patient in a continuous and generalized manner, in order to exhaust the immune system, which has proven to be useful in exhausting the T cells found in blood. On the other hand, there are multiple residence niches for T cells in peripheral organs, especially in the skin, intestines, liver, and lungs [67], which, in the case of the skin, have been shown to resist chemotherapy [68], which raises the possibility that this is not the only group of cells able to resist preventive treatment. Perhaps where this is most important is in hematopoietic stem cell transplantation, since this is the standard treatment for a large number of hematological diseases, where graft-versus-host disease is one of the most frequent causes of death. The main target organs of this pathology are the same peripheral organs and secondary lymphoid tissues; therefore, depleting the immune system through radiation, chemotherapy, or drugs seems to be the only possible option to avoid complications and improve graft survival. However, it has been shown that the depletion of memory T cells in bone marrow transplants can cause an increase in graft failure and the recurrence of leukemias and tumors [69,70]. It has now been found in clinical transplant trials that Treg inoculation manipulated by blocking the costimulatory pathway B7-CD28 using CTLA-4 Ig/Fc) or/and CD40-CD40L (using anti- CD154 mAb) demonstrated improved tolerance to transplants and reduced the need for immunosuppression [71].

## 8. Degenerative Diseases and Treg Cells

Degenerative diseases are generally associated with the aging process such as Alzheimer’s disease, multiple sclerosis, amyotrophic sclerosis (ALS), and Parkinson’s disease (PD) [72]. It is known that the development of these diseases is accompanied by a state of oxidative stress along with neuroinflammation and altered immune responses, which cause cell damage and death, and allow for the appearance of the disease as well as its progression. This loss of regulation of the immune response contributes to a vicious cycle between chronic oxidative stress and neuroinflammation. In addition, there is oxidation and the formation of extracellular protein aggregates, a mechanism similar to that used by prions [73]. The intracellular formation of free radicals can occur due to the influence of environmental factors such as ultraviolet radiation, ionizing radiation, and pollutants such as ozone, considered as prooxidant agents, produce ROS, which cause cell damage by generating an inflammatory response [74]. In a model of chronic exposure to low doses of ozone, it has been shown that it causes oxidative stress, in addition to a loss of regulation in the inflammatory response, which generates progressive neurodegeneration. Various studies using this model have shown that ozone induces mitochondrial dysfunction in the hippocampus [5], an increase in the expression of IL-17A in the hippocampus [75], changes in P2X7 receptors that promote various signaling pathways [3], and an overexpression of Syntaxin 5 in the hippocampus [76], in addition to generating cognitive deficits in object and place recognition tasks [77].

The neuroinflammation affects the central nervous system and is associated with the function of Teff cells and Treg cells in number and/or function. It has been reported that there is a neuroprotective response by Treg cells. There is evidence that demonstrates a relationship between CD3-activated Treg cells in mice exposed to 1-methyl-4-phenyl-1,2,3,6-tetrahydropyridine (MPTP) and a protection of the nigrostriatal system of more than 90% [78]. A higher number of CD4+/CD8+ T cells has also been found in the peripheral blood of patients with Alzheimer’s disease and in some mouse models [79]. In other degenerative diseases such as ALS, it was observed that they can have protective effects. In addition to the above, in a study with patients with ALS, it was found that the levels of Foxp3, TGF-β, IL4, and Gata3 mRNA decreased in those patients who progressed rapidly. In this study, both FoxP3 and Gata3 were accurate indicators of rates of progression [80].

The inhibitory role of Treg cells depends on cell contact-dependent inhibition through CD80/CD86 on APC cells, mainly dendritic cells, by the CTLA-4 receptor expressed on Treg cells [81]. A second suppression pathway occurs when there is a deprivation of IL-2 and IL-17 cytokines, generating cell death [82,83]. The physiologically required mechanism for the removal of amyloid β protein (Aβ) by the body is via insulin-degrading enzyme (IDE), a neutral zinc metallopeptidase responsible for the removal of amyloidogenic proteins to prevent amyloid formation [84]. However, when there is an exacerbated accumulation of Aβ, the microglia suffer a functional deterioration that facilitates neurotoxicity [85,86]. Specifically in Alzheimer’s disease, it has been shown that in addition to plaques and neurofibrillary tangles, chronic neuroinflammation is the third main pathological feature of Alzheimer’s disease, and is generated by the activation of microglia cells and the release of cytokines [87] such as TNF-α, a proinflammatory cytokine, which is elevated in patients with Alzheimer’s disease [88,89] as well as IL-6, IL-12, and IL-18, IFN-γ, chemokine (MCP-1), and other neurotoxic agents [90,91]. Another characteristic observed with AD is that the percentage of Treg cells and the levels of TGF-β in the blood of patients with mild cognitive impairment is higher in relation to patients with dementia. On the other hand, positive correlations were found between Treg cell percentage, IL-35, and cognitive assessment in patients with mild cognitive impairment and probable AD-related dementia [92].

## 9. Metabolic Diseases and Treg Cells

Loss of energy balance in cells can lead to the development of pathologies or metabolic disorders. According to the World Health Organization, many of these diseases, such as diabetes, atherosclerosis, obesity, and cardiovascular disease, and their complications are considered as a public health problem [93]. Inhalation of pollutants has been shown to contribute to metabolic disturbances [94] such as an increased risk of diabetes [95] and exacerbate preexisting cardiorespiratory conditions [96], among others. In a study with rats exposed to ozone (1 ppm), they observed that short and long chain free fatty acids rose uniformly after exposure to ozone [97], an alteration in the expression of certain genes involved in fatty acid metabolism and insulin signaling [98]. Exposure to ozone causes the activation of NF-κB in the lungs, in addition to an activation of cytokines and inflammatory proteins that contribute to the immune response [99], in addition to changes at the cellular level, generating epigenetic marks such as DNA methylation that promote inflammation and the development of diseases [100].

These diseases, characterized by presenting metabolic disorders and chronic inflammatory processes, mainly low-grade, are related to systemic metabolic deregulation, which creates an abnormal metabolic environment, promoting metabolic reprogramming [101]. The balance in the Th17/Treg response is regulated by inflammatory cytokines, metabolic factors, and epigenetic regulation [49]. In addition, excessive levels of ROS modify the structure and function of certain cellular proteins and lipids, which generates an alteration in energy metabolism, either in cell signaling, in the cell transport mechanism, and an activation of immune and inflammatory processes [102]. These alterations contribute to the development of metabolic diseases such as DM, which occur with a state of oxidative stress. In diabetes mellitus, there is a loss of immune regulation that generates pathogenic T cells and, therefore, the destruction of β cells in the islets of Langerhans. This process of glycolipotoxicity generates epigenetic acetylation-deacetylation signaling that regulates the functional activity of CD36 and subsequent lipid accumulation and caspase 3 activation in β-cells [103]. However, in DM, there is a deficiency of Treg cells in inflamed tissues, and this response is generated through the interleukin-2 (IL-2) receptor [104]. It has been shown that Treg cell replacement in this disease can reverse autoimmunity and protect β-cells [105]. On the other hand, FOXP3 is a factor that controls the development, transcriptional program, and suppressive function of Treg cells; therefore, CD4+CD25+FOXP3+ Treg cells are an essential immunosuppressive cell population for the control of homeostasis immune, the control of autoimmunity, and have a unique therapeutic profile. However, in DM, there is a deficiency of Treg cells in inflamed tissues, and this response is generated through the interleukin-2 (IL-2) receptor [104]. These abnormalities, produced by metabolic alterations, also produce hepatic inflammation, which is characterized by mitochondrial dysfunction through functional or structural alterations [106], perpetuating the damage. It has been established that the proinflammatory response generated by Th17 cells and the anti-inflammatory response of Treg cells maintain a balance, which serves to prevent excessive immune activation, autoimmune responses, and the pathogenesis of metabolic syndrome [49,107]. There are some cytokines, metabolites, and even microbiota that allow Th17 cells and Treg cells to be mutually linked and controlled by this microenvironment [108]. Glycolysis, glutaminolysis, and fatty acid metabolism are the three main metabolic pathways in CD4+ T cells that function to provide energy [109,110,111]. Activated T cells, therefore, undergo metabolic changes that are derived from metabolic reprogramming, with one of the first events being increased glycolysis to support biosynthesis and cell function [111]. Treg cells and memory T cells preferentially use oxidative phosphorylation as a source of ATP [112]. In addition, the changes caused by the migration of T cells increase the expression of the enzyme glucokinase, which, when associated with actin, promotes a rearrangement in the cytoskeleton in Treg cells [113]. Inflammasomes are intracellular complexes that, when associated with NOD-type receptors such as NLRP3, recruit pro-caspases 1, and a protein associated with apoptosis [114]. Some studies have suggested that IL-1β and IL-18 maturation and release is activated, and that this activation may be mediated through ROS production [115]. The inflammasome activation in hematopoietic cells alters insulin signaling in various tissues to reduce glucose tolerance and insulin sensitivity [116]. Some studies in NLRP3-deficient mice suggest a greater differentiation of the Treg cell population. This effect on CD4+ T cells downregulates the expression levels of Foxp3 translocating from the cytoplasm to the nucleus [117]. This suggests a form of negative regulation for the differentiation of Treg cells.

## 10. Antioxidants and Treg Cells

The activation and differentiation of Treg cells depends on signals from the microenvironment including antigens and cytokines as well as oxidative signals [118]. To maintain a balance in the production of ROS species, antioxidants play a very important role (Figure 7). During a chronic degenerative pathological event, prooxidants increase their production, establishing a state of chronic oxidative stress accompanied by inflammation, which produces a decrease in antioxidant defense systems.

### 10.1. Endogenous Antioxidants

Endogenous antioxidant systems maintain the redox balance in the body (Figure 7). The antioxidant enzyme copper/zinc superoxide dismutase (SOD-CuZn) is secreted by different types of cells, among which are thymic cells [119]. This secretion is carried out by micro-vesicles using ATP-dependent mechanisms [120]. Terrazzo G et al. demonstrated that antigen-dependent activation triggers the production and secretion of SOD-CuZn by human T cells in vitro; they also found that it is involved in T-cell receptor (TCR) signaling cascades [120], which is important for the differentiation of Treg cells [121]. SOD-CuZn dismutates the superoxide radical, transforming it into hydrogen peroxide, which is highly reactive. However, the enzyme glutathione peroxidase is responsible for transforming it into water and oxygen, thus preventing the oxidation of proteins, nucleic acids, and lipids. The enzyme phospholipid hydroperoxide glutathione peroxidase (GPx4) is responsible for reducing hydroperoxidated fatty acids [122]. Treg cells are known to depend on lipid metabolism for their survival and function, as they use extracellular free fatty acids to meet their metabolic demands [123,124], which increases the production of ROS intracellularly. Chengxian Xu et al. showed that Gpx4-dependent neutralization protects Treg cells from lipid peroxides, ferroptotic cell death, and in turn maintains Treg cell activation and function to control antitumor immunity. However, Gpx4 deficiency induces Treg cell ferroptosis, inducing a state of oxidative stress, which stimulates the production of proinflammatory cytokines such as IL-1β, increasing the inflammatory response [125]. Matsushita and collaborators observed that the addition of vitamin E to Gpx4-deficient T cells improved the survival of splenic T cells in vitro [126].

### 10.2. Exogenous Antioxidants

Exogenous antioxidants are those that we ingest in the diet such as vitamin C, which play a very important role in cell protection. Vitamin C is a molecule involved in different physiological functions, one of the most important is its role as a powerful antioxidant regulating the cellular redox state [125]. The administration of vitamin C to peripheral blood monocytes isolated from pneumonia patients has been shown to decrease the generation of the proinflammatory cytokines TNF- and IL-6 [126]. In addition, vitamin C is known to increase the activity of hypoxia-inducible factor 1α (HIF-1α) associated with oxidative stress [127], thus regulating the survival of T cells in humans [128] by promoting Th17 cell differentiation via increased IL-17 production, and regulating FoxP3 expression [129]. The differentiation and function of Treg cells depend on the expression of FOXP3. There are studies that show that the regulation of FOXP3 during the development of Alzheimer’s disease is very important, since the absence of this factor prevents the activation of Treg cells [130]. This is in addition to maintaining a balance between the TH17 cell and the Treg cells, since the loss of this balance has been involved in the development of autoimmune diseases and chronic diseases such as asthma. Studies by Jing-Guo Ma et al. 2021 showed that vitamin D protects asthmatic mice from airway inflammation by elevating Th17/Treg balance by inhibiting the NF-κB pathway [131]. Shelley Gorman et al., 2019 described a significant increase in the number of Treg cells in draining lymph nodes of the ear and skin, after treatment with 1,25(OH)2D [132]. Vitamin D also inhibits the proliferation of activated lymphocytes, reduces the production of inflammatory cytokines, and promotes the development of Treg cells. On the other hand, in the gastrointestinal tract, vitamin D inhibits Th17 cells and induces peripheral Treg cells to maintain the inflammatory balance [133]. In addition, vitamin D is essential to carry out the regulation of the immune system as well as other hormones such as melatonin (Table 1).

**Table 1 antioxidants-11-01553-t001:** The effect of endogenous and exogenous antioxidants on Treg cells.

Antioxidant	Animal Models/Cells	Mechanisms	References
**SOD-CuZn**	Human T cells in vitro	Its secretion by human T cells is involved in TCR signaling, important for Treg cell differentiation.	Gomez-Rodríguez J et al., 2014 [121]
**Gpx4**	Mouse strains C57BL/6 mice and Gpx4fl/fl mice	Gpx4-dependent lipid peroxide neutralization protects Treg cells from ferroptotic death and maintains Treg cell activation and function.	Chengxian Xu et al., 2021 [134]
**Vitamin C**	Renal carcinoma cells Foxp3-GFP mice C57BL/6 HIF-1αfl/fl mice Rorc^−/−^ mice	Increases the activity of hypoxia-inducible factor 1α (HIF-1α) and promotes the differentiation of Th17 cells, by regulating the expression of FoxP3.	Wohlrab C. et al., 2019 [129] Dang EV. et al., 2011 [131]
**Vitamin D**	Asthmatic mice	Protects from airway inflammation by elevating the Th17/Treg balance by inhibiting the NF-κB pathway.	Ma JG. et al., 2021 [133]
**Melatonin**	Necrotizing enterocolitis mouse	Blocks the differentiation of Th17 cells and increases the generation of Treg cells in vitro, by activating the AMPK/SIRT1 pathway in the intestine.	Ma F. et al., 2019 [135]
**Resveratrol**	Blood from asthmatic patients	Decreased phosphorylation of mTOR in Treg cells, decreased expression of IFN-γ in Treg cells, and increased expression of FoxP3.	Shimojima Y. et al., 2021 [136]
**Cyanidin**	Wistar rat model of rheumatoid arthritis	Inhibited the increase in Th17 cell differentiation and enhanced Treg cells.	Samarpita S. and Rasool. 2021 [137]

Melatonin is a hormone that is synthesized and secreted mainly by the pineal gland at night, and has a powerful antioxidant power and is involved in the modulation of the immune response [138]. It is known that in the intestinal inflammatory response, there is an imbalance between T helper 17 (Th17) cells and regulatory T cells, which favors the inflammatory process. Ma Fei et al., 2020 demonstrated that melatonin improved the intestinal balance of Th17/Treg cells in a mouse model of necrotizing enterocolitis [139]. They also showed that melatonin blocked the differentiation of Th17 cells and increased the generation of Treg cells in vitro, through the activation of the AMPK/SIRT1 pathway in the intestine. The role of antioxidants in the regulation of the immune system is essential to maintain an inflammatory balance. Each of them acts at different levels, activating or inhibiting different signaling pathways, which contribute in an orchestrated way to maintain both an oxidative and inflammatory balance. On the other hand, polyphenols are produced by plants as part of their metabolism and are involved in the protection, pigmentation, and reproduction of the plant. Among the polyphenols, we have the flavonoids, which are compounds that have antioxidant and anti-inflammatory properties [140]. Flavonoids are the most abundant polyphenols that are part of the human diet. These compounds have different metabolites such as resveratrol. Resveratrol is a flavonoid that has an antioxidant effect [141]. Resveratrol administration to patients with acute vasculitis contributes to a decrease in ROS production and a decrease in mTOR phosphorylation in Treg cells [136].

However, these results were also presented in the control patients. In addition to the above, they found a decrease in the expression of IFN-γ in Treg cells and an increase in the expression of FoxP3 after treatment with resveratrol in patients with acute vasculitis, however, the presence of Treg cells after treatment with resveratrol in patients, it was lower than in the control patients. It is worth mentioning that even when resveratrol failed to restore the expression of Treg cells, the results indicated that this compound had an important role as a modulator of the immune response. In addition to the above, cyanidin, which is a compound that is part of the anthocyanin family, was used in an experimental model of rheumatoid arthritis in Wistar rats by Samarpita S and Rasool in 2021 [137]. These authors demonstrated that cyanidin inhibited the increase in the differentiation of Th17 cells and improved Treg cells, which was observed both in vivo and in vitro. The results also indicated that cyanidin decreased IL-17 and increased IL-10 secretion in rats with arthritis. This demonstrates the importance of exogenous antioxidants and the role played by Treg cells in cell protection.

## 11. Discussion

It has been widely demonstrated that repeated exposure to low doses of ozone (similar to highly polluted days) causes a state of oxidative stress as repeated exposures to this gas are prolonged, which induces a state of oxidative stress in the body [5]. This loss of redox balance leads to a pro-oxidant change in the internal environment, in turn causing changes in redox signaling as well as in the intracellular signaling pathways of most systems in our body, for example, the immune system, metabolic, cardiovascular, nervous, etc. The aforementioned are manifested in the damage and evolution of non-infectious chronic degenerative diseases. However, as endogenous prooxidants and antioxidants, they modulate the immune system. The alteration of redox signaling leads the immune system to lose the regulation of its response, and the most notable manifestations are the increase in ROS, the failure of the antioxidant systems, and the loss of regulation of the inflammatory response [2], both being key factors in the deterioration that patients present as time goes by.

Treg cells play a crucial role in maintaining the regulation of both the immune system and the inflammatory response as they respond to changes in the microenvironment and regulate physiological responses such as during pregnancy and aging [52] as well as the fact that their alteration leads to different types of pathologies. On the other hand, the role of Treg cells is essential for fertilization, implantation, a successful pregnancy, and normal delivery to occur as well as in the control of the intestinal barrier and the normal aging process and other physiological processes. This is in addition to contributing during the redox balance to maintaining the regulation of the inflammatory response.

However, in the case of certain autoimmune diseases, the chronic state of oxidative stress is part of the environmental factors that contribute to altering tolerance, either by producing a decrease in Treg cells or by Treg cells losing their ability to regulate response [55], causing immunosuppression, which causes it to react with itself, producing autoimmunity, since both the central and peripheral tolerance processes are altered [58]. However, the chronic state of oxidative stress also causes the exhaustion of Treg cells, which implies that their proliferation ability is reduced and death cell is present. Therefore, this leads to a decrease in the immunosuppressive capacity of Treg cells [55].

In addition, the excess of ROS is capable of promoting the appearance of mutations that can favor tumor growth and, likewise, causes alterations in immune cells such as Treg cells, generating the suppression of the antitumor response; this suppression of the immune system appears in certain types of cancer [93]. It has been proposed that the suppression of the immune response depends on the endogenous antioxidant response of the Treg cells themselves that are present in the tumor and of the effector T lymphocytes [96]. On the other hand, an imbalance in the redox balance can also function as an aseptic activation mechanism of T cells, which conditions an acute or chronic rejection of the transplanted organ; therefore, these cells have become promising therapeutic targets that will reduce immunosuppression in post-transplant patients.

The chronic state of oxidative stress, neuroinflammation and immune alterations present in degenerative diseases cause Treg cells to alter in number and/or function, thus contributing to the loss of regulation of the inflammatory response. In addition, low-grade chronic inflammation also plays an important role in the development of metabolic diseases; therefore, the balance between the response of Th17/Treg cells is regulated by the levels of ROS, inflammatory cytokines, certain metabolic factors, and epigenetic control, which allow for metabolic reprogramming, protection against disease, or the activation of complex mechanisms such as the inflammasome, which regulates insulin resistance and inflammation.

The role of antioxidant systems when they lose their protective function is that the immune system alters the activation of Treg cells, which are responsible for maintaining the homeostasis of the inflammatory response; however, it has been observed that when endogenous (SOD-CuZn and GPX4) and exogenous (vitamins E, D, C and melatonin) antioxidants are administered, the Treg cells recover in number. Therefore, antioxidant systems play a crucial role in maintaining redox homeostasis.

In summary, we can assume that the type of response of Treg cells depends not only on the oxidation–reduction balance but also in part on the participation of the endogenous antioxidant systems of these cells in the pathophysiology of many diseases, since, as mentioned, in the case of certain tumors, the endogenous antioxidants produced by the Treg cells themselves protect the tumor from the effects of the immune system and allowed for the survival and growth of the tumor. In contrast, in certain autoimmune and neurodegenerative diseases, generalized oxidative stress can lead to a decrease in Treg cells, which changes the balance between proinflammatory and anti-inflammatory cytokines, maintaining the alteration in the immune response and the loss of regulation of the inflammatory response. which contributes to the deterioration and progression of these diseases over time. Therefore, the chronic state of oxidative stress may be a determining factor in the course of diseases and the pathways to regulate ROS may not only be effective targets, but also the endogenous antioxidant response of Treg cells that are infiltrated in certain type of tumors may represent another therapeutic target in the near future.

## 12. Conclusions

With the above, we can conclude the following: (1) The regulation in the number and function of Treg cells depends on the microenvironment in which the oxidation–reduction balance plays a central role. (2) In a redox balance, regulatory T cells play a fundamental physiological role in the tolerance of different barriers such as intestinal, blood–brain, lung, etc. (3) Furthermore, during physiological changes such as pregnancy, normal aging, etc., the participation of Treg cells allows these processes to be carried out normally. (4) Both exogenous and endogenous antioxidant systems play a fundamental role in the type of regulation (or loss of it) that T cells carry out on the immune system, and this may be an important fact that allows us to explain their different roles in different pathologies. (5) The chronic state of oxidative stress causes the exhaustion of Treg cells, which reduces their proliferation ability, leading to a decrease in the immunosuppressive capacity of Treg cells.

## Figures and Tables

**Figure 1 antioxidants-11-01553-f001:**
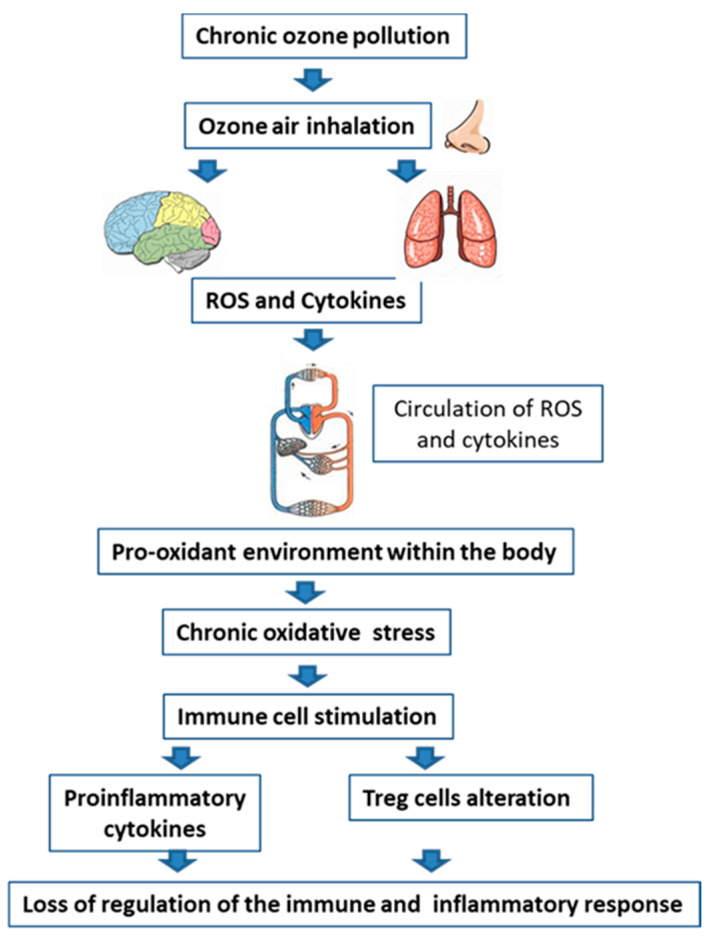
The effect of environmental pollution by ozone on the loss of regulation of the Inflammatory response. Chronic exposure to environmental pollution by ozone reaches the brain and lungs, causing an increase in the formation of ROS secondary to exposure to this gas. This induces an increase in prooxidants and cytokines, which leads to an increase in proinflammatory cytokines, causing a loss of regulation of the immune response in the body.

**Figure 2 antioxidants-11-01553-f002:**
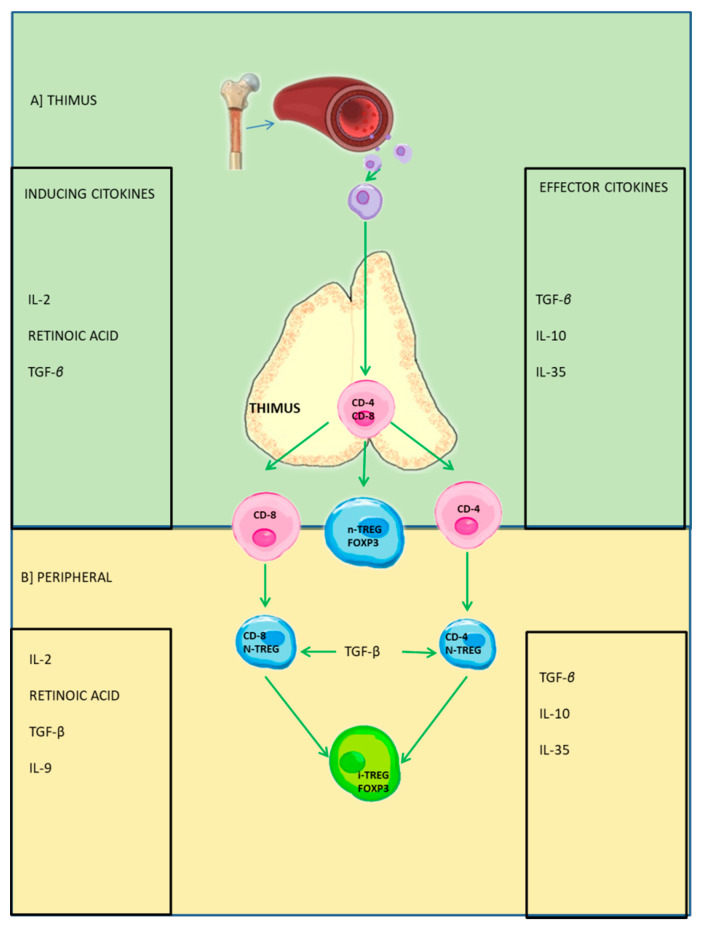
Treg cells: Thymic and peripheral activation. (**A**) Activation of Treg cells in the thymus via inducing cytokines and their effector cytokines. (**B**) Peripheral stimulation that promotes the activation of iTreg cells. The activation of these cells allows for balance in the immune response.

**Figure 3 antioxidants-11-01553-f003:**
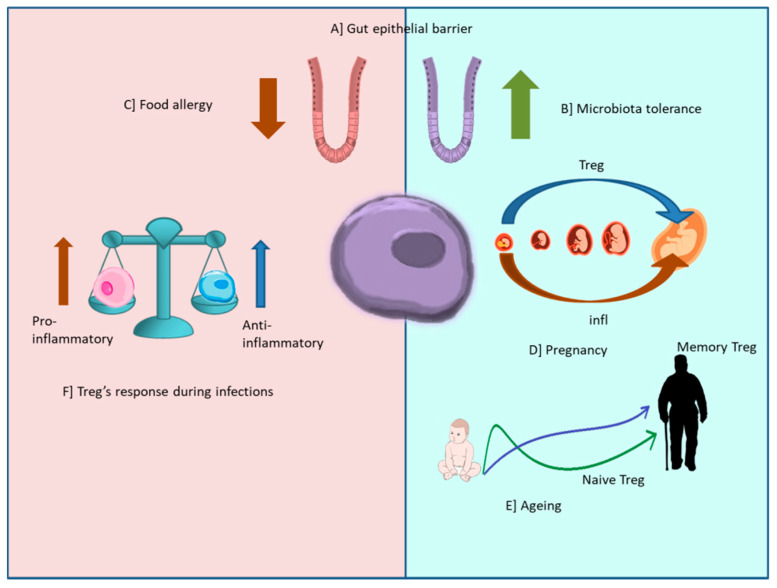
The mechanism of action of Treg during different physiological and pathological states (**A**). The figure shows the role of Treg cells in the intestinal epithelial barrier, the green arrow indicates the increase in Treg cells, which allows for immunological tolerance to the microbiota (**B**), the brown arrow represents the deregulation of Treg cells that leads to immune disorders such as celiac disease, food allergy, or irritable bowel (**C**). During pregnancy (**D**), the role of Treg is crucial for maternal–fetal tolerance and implantation, the purple arrow indicates that once fertilization occurs, a substantial increase in Treg cells is observed, creating a tolerogenic environment, whose composition changes throughout gestation; this increase decreases at the end and after each pregnancy. The red arrow indicates an inflammatory state at the beginning and at the end of gestation. During aging (**E**), the blue arrow indicates an increase in memory Treg cells and the green arrow indicates a decrease in naive Treg cells. During infections (**F**), the blue arrow indicates an increase in Treg cells that generates an anti-inflammatory state, while the predominance of Th17 cells, indicated by the red arrow, favors a pro-inflammatory state.

**Figure 4 antioxidants-11-01553-f004:**
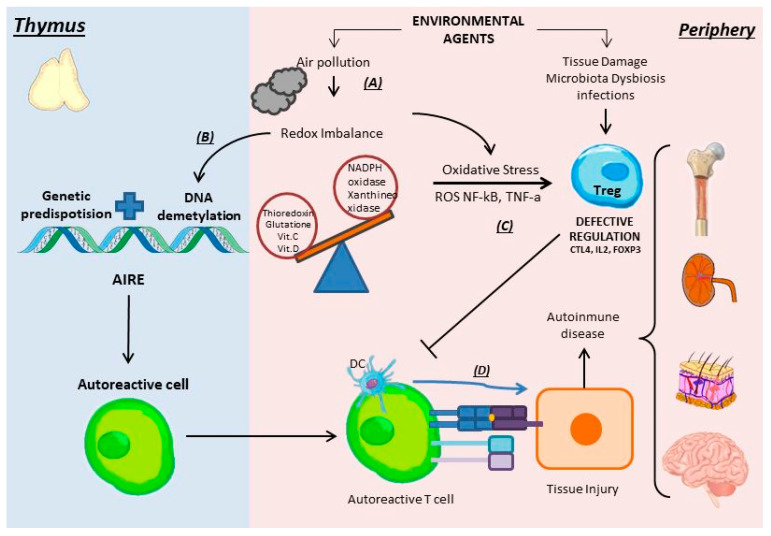
The redox imbalance in autoimmunity. (**A**) Environmental pollution induces a redox imbalance, with an increase in oxidizing enzymes (NADPH and xanthine oxidase) and a decrease in antioxidants (thioredoxin, glutathione, vitamins C, and D). (**B**) The redox imbalance generates DNA demethylation. (**C**) Oxidative stress (through ROS, NF-kB, TNF-α) modifies Tregs, reducing their regulatory capacity. (**D**) The autoreactive lymphocyte survives, proliferates, and damages the tissue generating autoimmune disease.

**Figure 5 antioxidants-11-01553-f005:**
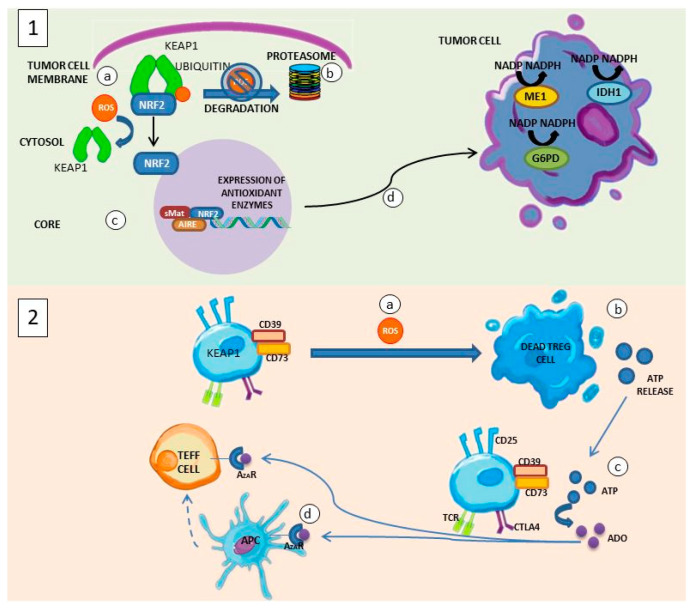
ROS and the tumor microenvironment. (**1**) Within the tumor microenvironment, tumor cells are able to survive by presenting oncogenic mutations such as that generated by Nrf2. (a) In the tumor cell cytosol, increased ROS levels prevent Nrf2 degradation mediated by Kelch-like ECH-associated protein 1 (KEAP1). (b) In the absence of ROS, Nrf2 will be degraded via the ubiquitin proteasome pathway. (c) When separated from KEAP1, Nrf2 translocates to the nucleus of the tumor cell where, together with Maf proteins, it binds to the antioxidant response element (ARE) and allows the expression of genes that code for the generation of antioxidant enzymes. (d) The tumor cell expresses NADPH-generating enzymes such as ME1, IDH1, and G6PD. (**2**) (a) Treg cells are susceptible to ROS-induced death. (b) A dead Treg cell will give way to the release of ATP. (c) ATP will be metabolized to adenosine (ADO) by CD39 AND CD73 of another Treg cell. (d) The generation of ADO is one of the suppression mechanisms of the antitumor immune response promoted by Treg cells because of the binding to the A2AR of effector T lymphocytes (Teff) and APCs as it generates immunosuppressive signals.

**Figure 6 antioxidants-11-01553-f006:**
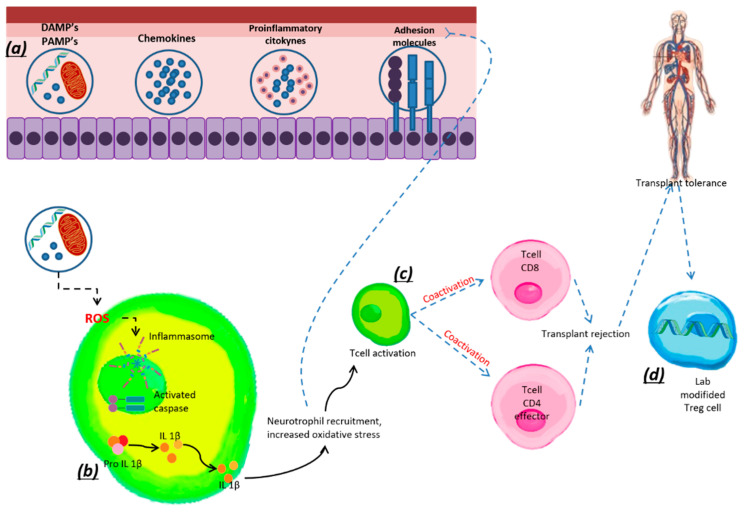
Pro-inflammatory mechanisms associated with the inflammation and activation of T cells. (**a**) PAMPs, DAMPs, chemokines, cytokines, and adhesion molecules that participate in inflammation (**b**) PAMPs and DAMPs, triggering a process of redox imbalance that favors the production of pro-inflammatory cytokines. (**c**) Aseptic activation of TCd8+ and TCd4+ cells, which leads to rejection of the transplanted organ. (**d**) Modified Treg cells induce tolerance to the transplant.

**Figure 7 antioxidants-11-01553-f007:**
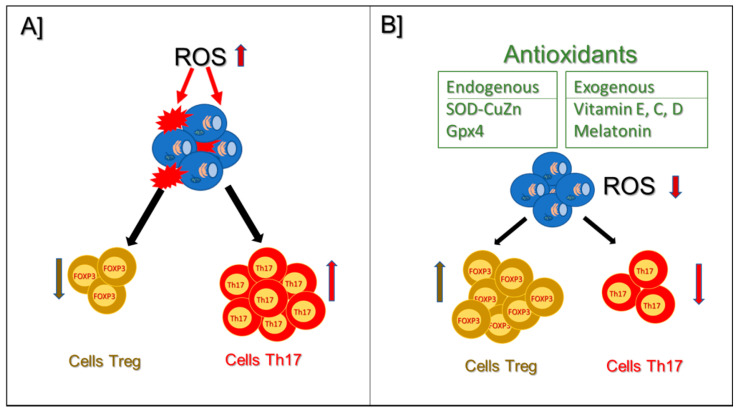
The role of antioxidants in maintaining the inflammatory balance between Treg cells and TH17 cells. (**A**) The figure shows that in a state of oxidative stress there is an imbalance between Treg cells (green arrow) and TH17 cells (red arrow). (**B**) In the presence of antioxidants (endogenous and exogenous), the number of Treg cells increases (green arrow), maintaining a compensatory balance to modulate inflammation.

## Data Availability

Not applicable.

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
