# Peer review of "Ozone Pollution, Oxidative Stress, Regulatory T Cells and Antioxidants"

_antioxidants, 2022, doi:10.3390/antiox11081553_

Round 1

Reviewer 1 Report

1. It is mentioned in the manuscript that ozone pollution can generate excessive free radicals and cause oxidative stress on T cells in the body and cause many diseases. However, the mechanism by which ozone generates reactive oxygen species is not clearly described. The authors should be added a graph to illustrate the association of ozone with excess reactive oxygen species and disease.

2. The author cites 6 papers that ozone produces oxidative stress to support the effect of ozone on T cells. The authors can add some other publications to illustrate that ozone can induce oxidative stress in T cells and lead to cell damage.

3. References 62 and 64 on page 8 both explain the effect of PM on T cells, but the author does not specify, why?

4. Page 16, why did the author only choose vitamin C and E, and melatonin among exogenous antioxidants to illustrate the antioxidant activity in T cells. Other natural antioxidants such as flavonoids and anthocyanins should also be discussed. The authors could be added an antioxidants table to illustrate the effect on T cells to be more convincing.

Author Response

Reviewer 1

First of all, I want to thank you for your suggestions that help us improve our work

  1. It is mentioned in the manuscript that ozone pollution can generate excessive free radicals and cause oxidative stress on T cells in the body and cause many diseases. However, the mechanism by which ozone generates reactive oxygen species is not clearly described. The authors should be added a graph to illustrate the association of ozone with excess reactive oxygen species and disease.

Response: The information was expanded, and the mechanisms by which ozone causes oxidative stress were described, and its association with the loss of regulation of the immune system.

  1. The author cites 6 papers that ozone produces oxidative stress to support the effect of ozone on T cells. The authors can add some other publications to illustrate that ozone can induce oxidative stress in T cells and lead to cell damage.

Response: This is a very interesting topic, which may be the key to understanding many chronic degenerative diseases, in which environmental pollution is playing a determining role. However, research has only just begun. Therefore, despite our search, there are very few articles on this topic.

  1. References 62 and 64 on page 8 both explain the effect of PM on T cells, but the author does not specify, why?

Response: This was clarified in the text by including contaminants in general

  1. Page 16, why did the author only choose vitamin C and E, and melatonin among exogenous antioxidants to illustrate the antioxidant activity in T cells. Other natural antioxidants such as flavonoids and anthocyanins should also be discussed. The authors could be added an antioxidants table to illustrate the effect on T cells to be more convincing.

Response: Other antioxidants such as flavonoids and anthocyanins were added. The requested table was also added.

Reviewer 2 Report

The paper Selva Rivas-Arancibia et al. “Ozone Pollution, Oxidative Stress, Regulatory T Cells And Antioxidants “ analyzed the possible correlation between ozone pollution exposure and regulatory cells

The paper is a review of the literature on T cells, in particular, Treg cells, and their implication in physiological and pathological conditions (pp4-14), while the title is focused on a more specific topic

The introduction specifies the topics to be investigated (Ozone, oxidative stress, antioxidants, T cells, and neurodegeneration), but then the authors get lost for about 10 pages explaining the function of the cells in all.

Example 1:, introduction lane 31 “… we have reported that exposure to low doses of ozone (0.25 parts 32 per million) in healthy rats causes alterations at the molecular, cellular and systemic levels, which, depending on the exposure time, produces a process of progressive neurodegeneration similar to what happens in neurodegenerative diseases such as Parkinson's  and Alzheimer's disease…”

But in the review this topic is sporadically investigated

Example 2: the word Ozone in the title appears in chapter 7.2 and is then appear only  10 pages later.

Example 3: Oxidative stress appears in chapter 7.2 and then sporadically about 10 times, but Ozone and Oxidative stress are keywords.

 Chapter 2 ozone pollution (lanes 62-82) makes no sense in that position and should be incorporated into environmental agents (7.2. Lane 306)

I recommend two proposals to the authors;

 1 change the title to something more generic (e.g., Function of cells in physiological and pathological conditions) and send it to another journal also MDPI (Biology, Biomedicine Immuno, IJMS)

 2. compact from pages 4 -14 and expand the part of Treg cells ozone and antioxidant (PP15-16).

Finally, in the review, many acronyms make reading very difficult for non-experts; I advise you to report the list of abbreviations on the first page.

Author Response

Reviewer 2

To Reviewer 2, I thank you for your time and suggestions to improve this writing,

The paper Selva Rivas-Arancibia et al. “Ozone Pollution, Oxidative Stress, Regulatory T Cells And Antioxidants “ analyzed the possible correlation between ozone pollution exposure and regulatory cells

1.-The paper is a review of the literature on T cells, in particular, Treg cells, and their implication in physiological and pathological conditions (pp4-14), while the title is focused on a more specific topic

Response: Deleted from page 2 to the top of page 4 and pages 6 and 7 to focus the article on the pathological changes when these cells are altered

2.- The introduction specifies the topics to be investigated (Ozone, oxidative stress, antioxidants, T cells, and neurodegeneration), but then the authors get lost for about 10 pages explaining the function of the cells in all.

Response: You are right, therefore in the introduction the effects of environmental pollution by ozone were deepened, and how it produces oxidative stress. Also, the effect of oxidative stress was also related to the response of Treg cells and the loss of regulation of the inflammatory response

Example 1:, introduction lane 31 “… we have reported that exposure to low doses of ozone (0.25 parts 32 per million) in healthy rats causes alterations at the molecular, cellular and systemic levels, which, depending on the exposure time, produces a process of progressive neurodegeneration similar to what happens in neurodegenerative diseases such as Parkinson's  and Alzheimer's disease…”But in the review this topic is sporadically investigated

Response: This topic was expanded

Example 2: the word Ozone in the title appears in chapter 7.2 and is then appear only  10 pages later.

Example 3: Oxidative stress appears in chapter 7.2 and then sporadically about 10 times, but Ozone and Oxidative stress are keywords.

Response: In this case, what we want to demonstrate in this work is how Treg cells are key in many pathological states, and also their association with chronic degenerative diseases. In addition, how the state of oxidative stress produced by environmental pollutants such as ozone, can be key to the development of non-infectious chronic diseases. As well as the role played by both endogenous and exogenous antioxidants in these cells

 Chapter 2 ozone pollution (lanes 62-82) makes no sense in that position and should be incorporated into environmental agents (7.2. Lane 306)

Response: Chapter 2 was rearranged and information was added

I recommend two proposals to the authors;

 1 change the title to something more generic (e.g., Function of cells in physiological and pathological conditions) and send it to another journal also MDPI (Biology, Biomedicine Immuno, IJMS)

  1. compact from pages 4 -14 and expand the part of Treg cells ozone and antioxidant (PP15-16).

Response: Some pages 2-3 and 6-7 were removed and the part of the ozone and antioxidant Treg cells was enlarged.

Finally, in the review, many acronyms make reading very difficult for non-experts; I advise you to report the list of abbreviations on the first page.

Response: A list of abbreviations will be made as requested

Round 2

Reviewer 2 Report

The paper in this revision is significantly improved and more focused on the topic of the title.

The added table is an excellent "summary," but I would ask the authors to insert a sentence at the end (or beginning) of chapter 10) example ... In table 1, are reported the principal effect of endogenous and exogenous antioxidants on Treg cells found in the literature." 

The work is accepted in this form